# Coarse-to-Fine Proposal Refinement Framework for Audio Temporal Forgery Detection and Localization

Junyan Wu
wujy298@mail2.sysu.edu.cn
School of Computer Science and
Engineering, Sun Yat-sen University
Guangzhou, China

Wei Lu*
luwei3@mail.sysu.edu.cn
School of Computer Science and
Engineering, Sun Yat-sen University
Guangzhou, China

Xiangyang Luo
luoxy_ieu@sina.com
State Key Laboratory of Mathematical
Engineering and Advanced
Computing, Zhengzhou, China

Rui Yang
duming.yr@alibaba-inc.com
Alibaba Group
Hangzhou, China

Qian Wang
qianwang@whu.edu.cn
School of Cyber Science and
Engineering, Wuhan University
Wuhan, China

Xiaochun Cao
caoxiaochun@mail.sysu.edu.cn
School of Cyber Science and
Technology, Shenzhen Campus of Sun
Yat-sen University, Shenzhen, China

## Abstract

Recently, a novel form of audio partial forgery has posed challenges to its forensics, requiring advanced countermeasures to detect subtle forgery manipulations within long-duration audio. However, existing countermeasures still serve a classification purpose and fail to perform meaningful analysis of the start and end timestamps of partial forgery segments. To address this challenge, we introduce a novel coarse-to-fine proposal refinement framework (CFPRF) that incorporates a frame-level detection network (FDN) and a proposal refinement network (PRN) for audio temporal forgery detection and localization. Specifically, the FDN aims to mine informative inconsistency cues between real and fake frames to obtain discriminative features that are beneficial for roughly indicating forgery regions. The PRN is responsible for predicting confidence scores and regression offsets to refine the coarse-grained proposals derived from the FDN. To learn robust discriminative features, we devise a difference-aware feature learning (DAFL) module guided by contrastive representation learning to enlarge the sensitive differences between different frames induced by minor manipulations. We further design a boundary-aware feature enhancement (BAFE) module to capture the contextual information of multiple transition boundaries and guide the interaction between boundary information and temporal features via a cross-attention mechanism. Extensive experiments show that our CFPRF achieves state-of-the-art performance on various datasets, including LAV-DF, ASVS2019PS, and HAD.

## CCS Concepts

• **Computing methodologies → Artificial intelligence**; **Computer vision**; **Computer vision problems**;

*Corresponding author.

## Keywords

Audio forensics, temporal forgery localization, partial forgery detection

**ACM Reference Format:**
Junyan Wu, Wei Lu, Xiangyang Luo, Rui Yang, Qian Wang, and Xiaochun Cao. 2024. Coarse-to-Fine Proposal Refinement Framework for Audio Temporal Forgery Detection and Localization. In *Proceedings of the 32nd ACM International Conference on Multimedia (MM '24), October 28-November 1, 2024, Melbourne, VIC, Australia.* ACM, New York, NY, USA, 9 pages. https://doi.org/10.1145/3664647.3680585

## 1 Introduction

Driven by massive amounts of data and powerful deep learning networks [9–12, 17, 24], lifelike productions of audio artificial intelligence generated content (AIGC) [2, 14–16, 22, 23] have become more accessible to the public. While AIGC-related industries have greatly improved the quality of people's lives, they also bring convenience to malicious users. Currently, a novel form of audio partial forgery using AIGC and a large language model has raised public concerns about the security and authenticity of network information. Figure 1 illustrates the process of fabricating disinformation that AIGC productions can be integrated seamlessly into the original audio by minor modifications. Unfortunately, current countermeasures mainly address the entire synthesized utterances [1, 13, 26, 28, 40], leaving the door open for malicious users to conceal forgery traces at a low cost. To promote the safe and proper use of AIGC, it is significant to study a more accurate and reliable countermeasure against partial forgery audio.

Recent research efforts have been devoted to creating audio partial forgery datasets [4, 7, 34, 35, 37] and proposed a challenging task known as audio partial forgery detection (PFD). As shown in the 'countermeasure' part of Figure 1, the PFD task aims to detect partial forgery segments, which may be only a small portion of the real audio, at different detection resolutions. Two main solutions have been proposed for the audio PFD, including forgery content detection [18, 31, 33, 37, 41] and forgery boundary detection [6, 8]. The former primarily focus on determining the content authenticity, with the goal of distinguishing which segments within audio are real or forgeries. The latter focus on detecting the transition boundary between real and forgery segments. Among them,

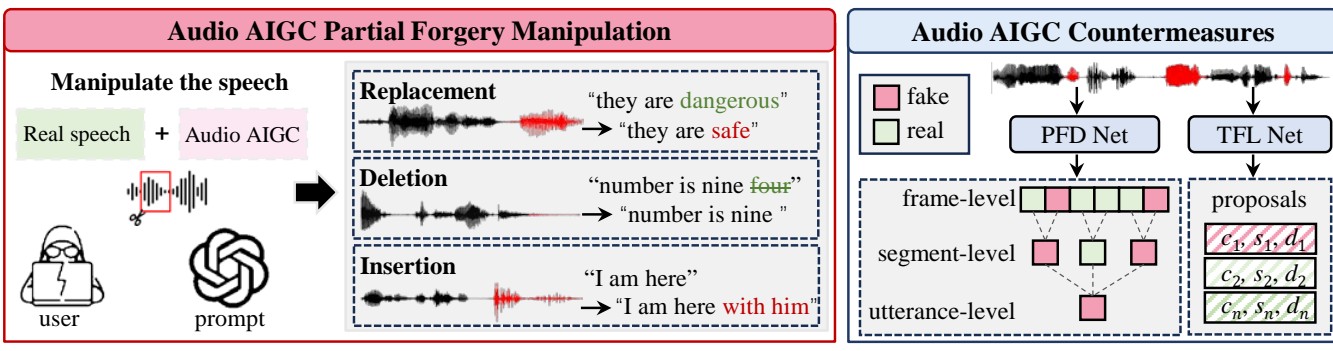

**Figure 1: The schematic diagram of a novel partial audio forgery and its countermeasures: the partial forgery detection (PFD) network identifies the content authenticity at different detection resolutions, while the temporal forgery localization (TFL) network predicts proposal regions (confidence score, start timestamp, duration length) for forgery segments.**

self-supervised learning (SSL) models play a significant role in the detection performance improvement. The basic idea is to leverage powerful audio representations from the pre-trained SSL model to facilitate downstream forgery detection tasks. Despite the remarkable advancement, there still exist some issues.

**(1) Classification limitation:** Current audio PFD solutions still aim at classification, i.e., ranging from predicting utterance-level results to frame-level results. However, providing temporal forgery regions within the modified audio can help users better understand the analysis results of the audio forgery content, which aligns more closely with the practical needs and applications of audio forensics. To fill the gap in audio forensics, we introduce a temporal forgery localization (TFL) task, which not only identifies each audio segment as real or forgery but also determines the precise timestamps at which these forgery segments start and end.

**(2) Small forgery segment challenge:** If malicious users have phonological knowledge, they can alter the original semantics by manipulating vowels and even consonants smaller than the word level. TFL network aims to predict specific forgery regions, and it may be challenging to locate small forgery segments consisting of a single frame or several consecutive frames in long-duration audio. Thus, it is desirable to improve the localization performance of small forgery segments by embedding prior information of frame-level detection scores from the PFD network into the TFL network.

**(3) Multiple forgery segment challenge:** Malicious users may not simply manipulate a single segment within audio but deliberately employ forgery in multiple segments. This challenge causes localization performance to gradually weaken with the increasing number of forgery segments. The transition boundary can provide valuable cues to enhance the detection of multiple forgery segments, as it indicates artifacts like inconsistency in speech and inconsistency in ambient noise [30]. Therefore, it is vital to introduce the contextual information at forgery transition boundaries.

In this work, we are motivated to propose a novel two-stage framework called the coarse-to-fine proposal refinement framework (CFPRF) [1] for audio temporal forgery detection and localization (TFDL). Unlike existing audio PFD methods, we leverage a frame-level detection network (FDN) in the first stage to learn

robust representations for better indicating rough forgery regions and employ a proposal refinement network (PRN) in the second stage to produce fine-grained proposals. To address the audio TFDL challenges, our core idea is to mine temporal inconsistency cues by forcing the model to perceive the subtle differences between different frames and to capture the contextual information of multiple transition boundaries. To enlarge the sensitive differences caused by small forgery segments, we employ contrastive representation learning (CRL) [33] in the difference-aware feature learning (DAFL) module to guide distinct frame pairs to exhibit dissimilarity in the embedding space while ensuring identical frame pairs demonstrate similarity. In addition, we design a boundary-aware feature enhancement (BAFE) to guide the interaction between boundary context information and temporal difference-aware features to enhance the detection of multiple forgery segments. Benefiting from these modules, the robust frame-level features can be learned to provide pivotal coarse-grained proposals, which are conducive to locating small and multiple forgery segments and further being refined in the PRN. The results demonstrate the effectiveness of our CFPRF and the practicality of the designed PRN. Overall, the main contributions of this paper can be concluded as follows:

- We innovatively propose a two-stage framework called CF-PRF, including an FDN for frame-level forgery detection and a PRN for temporal forgery localization.
- We introduce a DAFL module involving dual-attention layers and guided by CRL to enlarge the sensitive differences between different frames caused by minor manipulations.
- We devise a BAFE module incorporating the cross-attention mechanism to enhance the temporal features with context information of multiple transition boundaries.
- We present a plug-and-play PRN to predict confidence scores and regression offsets to refine the coarse-grained proposals derived from the FDN. Moreover, PRN can be integrated with existing audio PFD models for localization.

## 2 Related Work

### 2.1 Audio Forgery Detection

The audio forgery detection task aims to determine whether an audio contains forgery content. Early audio forgery detection methods

[1]Code and pre-trained models are available at: https://github.com/ItzJuny/CFPRF.

[1, 13, 26, 28, 40] focus on introducing a well-designed model to capture forgery traces from a global view to detect entirely synthesized utterances. For example, AASIST [13] introduced a heterogeneous graph fusion network to model the audio temporal-spectral subgraphs, which are then downsampled to utterance-level scores. In addition, ASDG [32] utilized aggregation and separation domain generalization to learn an optimal feature space that can aggregate real audio and fake audio, resulting in better generalizability in detecting unseen target domains. Although cheerful performance has been achieved in various utterance-level detection scenarios, an initial investigation into audio partial forgery detection [38] revealed limitations. PFD task requires finer detection resolutions to identify subtle manipulations that affect only a small portion of the original audio. In this regard, researchers have explored PFD models under various time resolutions [6, 8, 18, 31, 33, 37, 41], ranging from longer segment-level resolution (e.g., 320 milliseconds) down to frame-level one (20 milliseconds). For instance, QASAM [31] leveraged the question-answering strategy and self-attention mechanism into a designed fake span detection module, directing the model to focus away from less generalization shortcuts into fake spans. LS-HMI [41] used a hybrid multi-instance learning to mitigate the gradient conflict problem and applied local self-attention to compute local correlations between segments from a temporal perspective, thus enhancing feature representation. In addition, PSDL [37] and MGBF[18] implemented an SSL front-end model and combined multiple detection resolutions to improve the flexibility and accuracy of detection. TDL [33] was drew on the SSL front-end and designed an embedding similarity module to separate real and fake frames in the feature space. Moreover, WBD [8] and IFBDN [6] also introduced an SSL model and applied transform blocks to detect the probability of being a boundary for each frame.

However, existing audio PFD methods have classification limitations that can not specify the start and end timestamps of partial forgery segments, thus limiting the practicality of audio forensics in realistic scenarios. To overcome this limitation, the temporal forgery localization task was introduced.

## 2.2 Temporal Forgery Localization

Recently, temporal forgery localization has received considerable attention in video forensics. It aims to acquire informative visual-audio representations by exploring the interaction between multi-modal features and to integrate an established localization decoder [19, 25, 36] to realize the localization of partial forgery segments. LAV-DF [7] was the first to introduce a content-driven multi-modal TFL dataset and proposed a benchmark network BA-TFD guided via contrastive learning, frame classification, and boundary matching network (BMN) [19] for the TFL task. In their subsequent work, BA-TFD+ [5] enhanced the BA-TFD backbone with a multi-scale transformer and adopted a BSN++ [25]-based boundary module to improve the localization performance significantly. In addition, AV-TFD [21] also drew on the decoder structure of BMN and used a cross-modal attention mechanism and an embedding-level fusion mechanism for learning visual-audio representations. To improve the localization accuracy of the model, UMMAFormer [39] devised a temporal feature abnormal attention module and a parallel cross-attention feature pyramid network for enhancing subtle features and incorporated an ActionFormer decoder for localization[36].

However, we observe that existing vision-audio TFL methods cannot be directly applied to the audio TFL task. The forgery video usually manipulates successive video frames to effect semantic changes, while the forgery audio alters meaning with just a vowel change (single frame). In this context, these end-to-end TFL methods may not address small- and multi- forgery segment challenges in audio scenarios. Moreover, current audio forgery boundary detection models are unable to determine the start and end timestamps of partial forgery segments. In light of these issues, we propose a solution called CFPRF for audio TFDL.

## 2.3 Task Definition

Given a partial forgery audio, we aim to detect partial forgeries at the frame level and propose the locations of these forgery segments.

*2.3.1 Frame-level Forgery Detection.* Take the input audio $x$ as an example, which contains $T$ frames. First, frame-level features are learned by a designed encoder, denoted as $F_x \in \mathbb{R}^{T \times D}$. Then, $F_x$ is decoded into forgery probability scores $\hat{Y}_x = \{\hat{y}_1, \hat{y}_2, ..., \hat{y}_T\}$. This task can be considered as a frame-level classification challenge, utilizing labels $Y_x = \{y_1, y_2, ..., y_T\}$ where 0 denotes a fake frame and 1 denotes a real frame, respectively.

*2.3.2 Temporal Forgery Localization.* Each audio $x$ is associated with a set of temporal forgery ground truths $G_x = \{(s_m, d_m)\}_{m=1}^M$, where $M$ is the number of forgery segments within the audio. For the $m$-th forgery segment, $s_m$ and $d_m$ are the start timestamp and duration time, respectively. This task needs to predict a set of confidence scores $C_x = \{c_1, c_2, ..., c_H\}$ for each coarse-grained proposal $P_x^\dagger = \{(s_h^\dagger, d_h^\dagger)\}_{h=1}^H$ derived from FDN outputs, where $H$ is the number of coarse-grained proposals. Then, positive proposals are refined using predicted regression offsets $\hat{R}_x = \{(\hat{r}_h^s, \hat{r}_h^d)\}_{h=1}^{H_{pos}}$, where $H_{pos}$ is the number of positive proposals, each with a confidence score $c_h \geq \theta_p$. For the $h$-th proposal, labels for the duration offset $r_h^d$ ($r_h^d = log(d_h/d_h^\dagger)$) and the shift offset $r_h^s$ ($r_h^s = (s_h - s_h^\dagger)/d_h$) are calculated based on matched ground truth.

## 3 Proposed Method

## 3.1 Frame-level Detection Network

To learn robust discriminative features, we first design a frame-level detection network to mine inconsistency cues to better distinguish between fake and real audio frames. Figure 2 illustrates the detail of our FDN, which contains difference-aware feature learning and boundary-aware feature enhancement modules that are adept at capturing minute forgery traces. As the SSL model (namely, Wav2Vec2-XLSR300M) has been shown to produce powerful audio representations for downstream tasks, we first extract front-end features from the last hidden states using an embedding size of 1024. To reduce feature dimension, we apply a linear layer to get the output feature map $F_{ssl} \in \mathbb{R}^{T \times 128}$, where $T$ is the number of audio frames. These features are then fine-tuned to be tailored for the PFD task. Considering features $F_{ssl}$ are not enough for modeling spectral and channel-wise information for each frame, we utilize six CNN-based residual blocks to learn a higher-level feature map $F_{sc} \in \mathbb{R}^{C \times T \times S}$. Here, $C$ and $S$ are the number of channels and spectral bins for each frame, respectively.

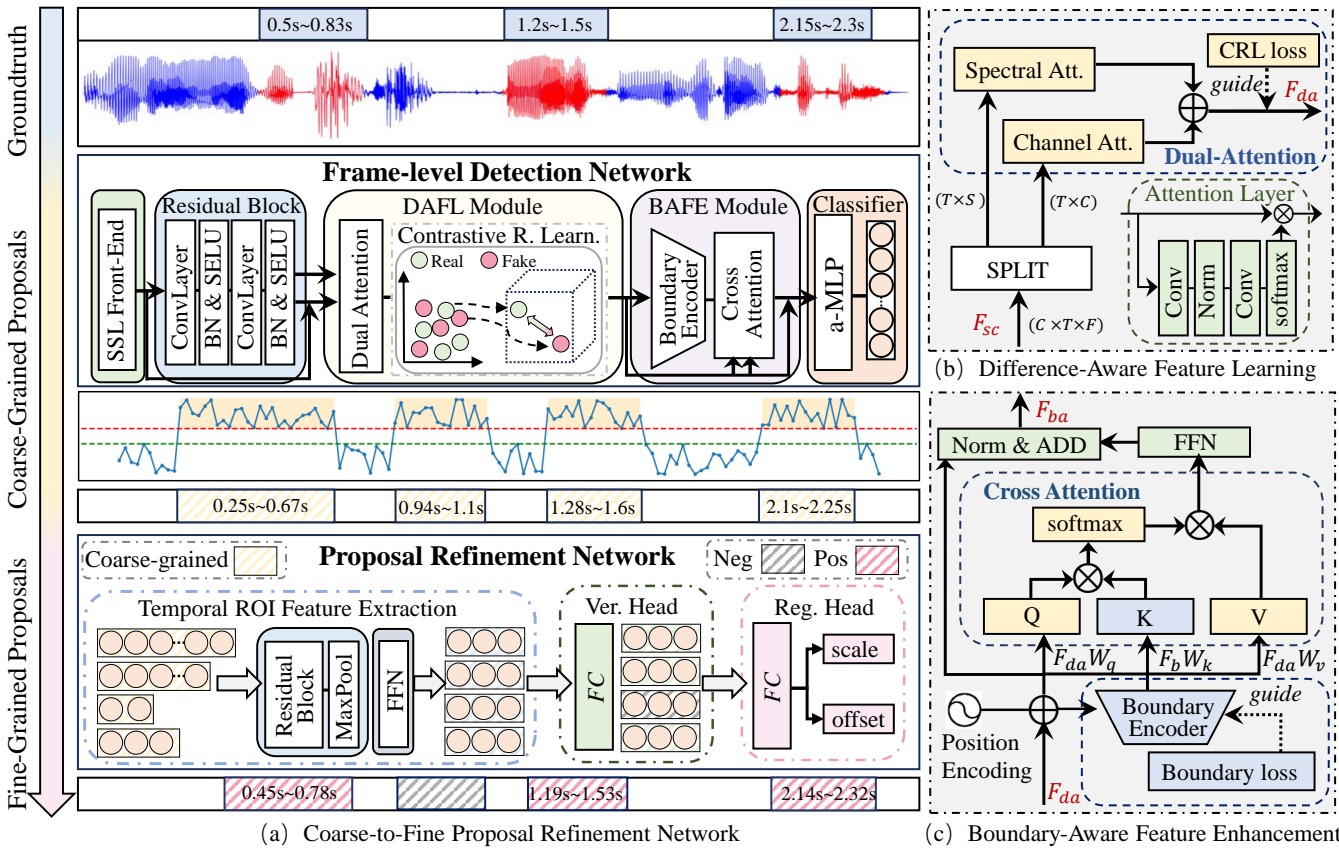

Figure 2: The structure of the proposed coarse-to-fine proposal refinement framework (CFPRF), involving a frame-level detection network and a proposal refinement network for audio temporal forgery detection and localization.

**Difference-aware Feature Learning Module.** Figure 2(b) shows the structure of DAFL involving dual-attention layers and guided by contrastive representation learning to enlarge the sensitive differences between different frames caused by minor manipulations. To better focus on either spectral or channel information, we separate $F_{sc}$ to construct spectral representation $F_s \in \mathbb{R}^{T \times S}$ and channel representation $F_c \in \mathbb{R}^{T \times C}$. Then, dual attention layers are applied to derive attention weight matrices $M_c$ and $M_s$ across the spectral and the channel dimension, respectively, which can be denoted as

$$M_s = \text{softmax}(E_{al}(F_s)),$$
$$M_c = \text{softmax}(E_{al}(F_c)), \quad (1)$$

where $E_{al}(\cdot)$ denotes the encoder of the attention layer, followed by a softmax function to normalize these weights. The integration of spectral-channel information has better potential to leverage complementary information to model temporal forgery representations, which can be denoted as

$$F_t = (F_c \odot M_c) \oplus (F_s \odot M_s), \quad (2)$$

where $F_t \in \mathbb{R}^{T \times D}$ ($D = C + S$) is the temporal representation, $\odot$ denotes the element-wise multiplication, and $\oplus$ indicates the element-wise addition.

To enlarge the differences between real and fake frames in the feature space, we further enable a pair of real and fake frames from different positions to exhibit dissimilarity. In contrast, a pair of frames that are both either real or fake should show similarity. Using cosine similarity to measure the similarity between feature vectors is a popular approach. Specifically, the cosine similarity between two frames $f_a$ and $f_b$ can be denoted as

$$SIM(f_a, f_b) = \frac{f_a^T \cdot f_b}{\| f_a \|_2 \cdot \| f_b \|_2} \quad (3)$$

Then, the contrastive learning loss $L_r$ is utilized to guide difference-aware features $F_{da}$ to cluster similar frames together while separating dissimilar frames, denoted as

$$L_r = \frac{1}{J} \sum_{j=1}^{J} I_j (1 - SIM(f_j, f_j^+))^2 + (1 - I_j) max(0, SIM(f_j, f_j^-) - \alpha)^2 \quad (4)$$

where $J$ is the number of frame pairs. For the $j$-th pair, $f_j$ is the feature vector of the reference frame, while $f_j^+$ and $f_j^-$ are the feature vectors of the similar and dissimilar frames, respectively. $I_j$ set to 1 is used to indicate similar pairs and 0 otherwise, while $\alpha$ is a margin parameter for the dissimilar pairs.

**Boundary-aware Feature Enhancement Module.** The transition boundaries of partial forgery segments are often presented unnaturally, such as discontinuities in speech and inconsistencies in ambient noise[30], leaving the door open for detection. Inspired by this, we design a boundary-aware feature enhancement module, as shown in Figure 2(c), with the goal of enhancing features by modeling context information of multiple transition boundaries. To model long-term information and capture boundary inconsistencies, a boundary encoder based on MLPs with gating and a tiny self-attention (aMLP) [20] is utilized to get the boundary feature map $F_b$. Further, a full connected layer is applied to downsample $F_b$ to boundary probability scores $\hat{Y}_b = \{\hat{y}_b^1, \hat{y}_b^2, ..., \hat{y}_b^T\}$. Considering the imbalance problem in boundary labels, we employ a hyperparameter-free loss function, namely P2sGrad-based mean squared error (MSE) [29], which can be defined as follows:

$$L_b = \frac{1}{T} \sum_{t=1}^{T} MSE_{pg}(\hat{y}_b^t, y_b^t) \tag{5}$$

where $MSE_{pg}(\cdot)$ is the P2sGrad-based MSE loss, $T$ is the number of audio frames. Each frame is assigned a boundary label, where $y_b^t$ is 1 denoting the $t$-th frame as a transition point, and 0 otherwise. For enhancing the temporal difference-aware features, we employ a cross-attention transformer block to facilitate the interaction between boundary information and temporal features. Given the permutation-invariant problem, the positional encoding $PE(\cdot)$ is added to the inputs to provide the positional information. Within this block, the boundary-aware correlation map $M_{ba}$ between temporal features $F_{da}$ and boundary features $F_b$ are first derived and normalized as follows:

$$Q = W_q(PE(F_{da})), K = W_k(PE(F_b)),$$
$$M_{ba} = \text{softmax}(\frac{QK^T}{\sqrt{C}}), \tag{6}$$

where learning matrices $\{W_q, W_k\} \in \mathbb{R}^{D \times D'}$ is utilized to transform features $\{F_{da}, F_b\}$, and $\frac{1}{\sqrt{C}}$ dentoes the scaling factor. To get the output features enhanced by the boundary-aware attention, a dot-production is performed on the temporal features projected by $W_v \in \mathbb{R}^{D \times D'}$ and the cross-attention map $M_{ba}$, which can be denoted as

$$V = W_v(PE(F_{da})), F_{ca} = M_{ba}V \tag{7}$$

Additionally, the enhanced features $F_{ca}$ are added to the original features $F_{da}$ and fed to a feed-forward network with layer normalization to generate boundary-aware representations $F_{ba}$. Finally, an aMLP-based decoder is utilized to downsample $F_{ba}$ to frame-level forgery probability scores $\hat{Y}_f = \{\hat{y}_f^1, \hat{y}_f^2, ..., \hat{y}_f^T\}$. Also, we employ the P2sGrad-based MSE function to compute the forgery detection loss $L_f$ as

$$L_f = \frac{1}{T} \sum_{t=1}^{T} MSE_{pg}(\hat{y}_f^t, y_f^t), \tag{8}$$

where $T$ is the number of audio frames, and label $y_f^t$ is 0 denoting the $t$-th frame as a forgery frame, and 1 otherwise.

## 3.2 Proposal Refinement Network

To realize the localization of partial forgery segments, we devise a proposal refinement network in the second stage, whose structure

is shown at the bottom of Figure 2(a). We utilize the frame-level features and detection scores derived from the PFD network to better locate small and multiple forgery segments in long-duration audio. The frame-level detection scores are used to provide coarse regions, and region features are utilized to predict confidence scores and regression offsets to produce fine-grained proposals. Following a two-step strategy, we fix the network parameters of the FDN to train the PRN.

**Coarse-grained Proposal.** To be specific, PRN takes frame-level forgery probability scores $\hat{Y}_f = \{\hat{y}_f^t\}_{t=1}^{T}$ and the last hidden states $F_{ba} = \{f_{ba}^t\}_{t=1}^{T}$ derived from the FDN as input, where $\hat{y}_f^t$ and $f_{ba}^h$ denote the score and the feature vector of the $h$-th frame, respectively. Given predicted forgery scores, a temporal region $(s, d)$ that likely to contain forgeries is formed by consecutive forgery frames $\{(\hat{y}_f^t)|\hat{y}_f^t > \theta_f\}_{t=s}^{s+d}$, each forgery frame with a score higher than a threshold $\theta_f$. In this context, $s$ and $d$ are the start timestamp and the duration length of the region, respectively. Furthermore, coarse-grained proposals $P^{\dagger} = \{(s_h, d_h)\}_{h=1}^{H}$ can be defined as a set of pairs containing the start timestamp and the duration length, where $H$ is the number of coarse-grained proposals. For the $h$-th proposal, the corresponding region features can be extracted as $R(P_h^{\dagger}) = \{f_{ba}^h\}_{h=s_h}^{s_h+d_h}$. To enhance the incorporation of global-context information for subsequent validation and regression, we design a pooling encoder consisting of two CNN-based residual blocks and a max-pooling layer, where each residual block contains two convolutional layers followed by BN and SELU functions. Consequently, the pooling encoder processes coarse-grained region features and outputs TRoI features $F_r \in \mathbb{R}^{H \times D'}$ with a fixed size.

**Fine-grained Proposal.** PRN is responsible for producing the fine-grained proposals and the confidence scores. To this end, two MLP-based headers are employed that take TRoI features $F_r$ as input to decide whether a proposal is positive and refine these positive proposals using predicted regression offsets. We define three types of proposals based on the temporal intersection-over-union (TIoU) with their matched groundtruths: mostly-complete proposals with $TIoU > 0.7$, incomplete proposals with $0.4 <= TIoU < 0.7$, and otherwise negative proposals. Then, a binary cross-entropy loss is adopted to derive the verification loss $L_v$, which can be denoted as follows:

$$L_v = -\frac{1}{H} \sum_{h=1}^{H} \left( y_v^h \log(\hat{y}_v^h) + (1 - y_v^h) \log(1 - \hat{y}_v^h) \right) \tag{9}$$

where $y_v^h$ is the true label for the $h$-th proposal, with $y_v^h = 1$ indicating a positive proposal and 0 otherwise.

In addition, a regression header is crucial for precisely approaching positive proposals to their groundtruths. Consequently, $F_r$ within $H_{pos}$ positive proposals are fed into the regression header to obtain regression offsets $\hat{R} = \{(\hat{r}_h^s, \hat{r}_h^d)\}_{h=1}^{H_{pos}}$, including predicted start offsets and duration offsets. The regression loss is calculated only for positive proposals and can be expressed as

$$L_r = \frac{1}{H_{pos}} \sum_{h=1}^{H_{pos}} y_v^h |(s_h - \hat{s}_h) + (d_h - \hat{d}_h)| \tag{10}$$

where $H_{pos}$ is the number of positive samples, $s_h$ and $d_h$ are the true labels of the start offset and the duration offset for the $h$-th proposal, respectively. In this context, Soomth-L1 distance is adopted, where $y_v^h = 1$ indicates the positive proposal and 0 otherwise.

Finally, fine-grained proposals $P = \{(s_h + \hat{s}_h, d_i + \hat{d}_h)\}_{h=1}^{H_{pos}}$ are generated. Ideally, our proposed PRN can be integrated with other PFD networks to achieve temporal forgery localization using similar processes upon frame-level features and forgery scores.

## 3.3 Training and Inference

**Training.** We utilize a two-step strategy to train the CFPRF: the FDN is first trained on the PFD task, and then its network parameters are fixed to conduct the subsequent the PRN training. The training of the FDN is guided by the CRL loss $L_c$, the forgery detection loss $L_f$, and the boundary detection loss $L_b$. And the total loss $L_{FDN}$ for training FDN is calculated as

$$L_{FDN} = L_f + \lambda_c * L_c + \lambda_b * L_b, \quad (11)$$

where $\lambda_c = 0.15$ and $\lambda_b = 0.1$ are hyper-parameters to balance the total loss. In addition, the total loss $L_{PRN}$ for training the PRN is calculated as

$$L_{PRN} = L_v + \lambda_r * L_r, \quad (12)$$

where $L_v$ and $L_r$ are the verification loss and the regression loss, respectively. And the hyper-parameter $\lambda_r$ is set to 0.15. Given that coarse-grained proposals derived from forgery probability scores are mostly composed of positive proposals, causing an imbalance problem in training the verification header. To ensure stable training, we randomly sample negative proposals to balance the number of negative and positive proposals.

**Inference.** For inference, we utilize the FDN to generate forgery probability scores $\hat{Y}_f$, and then apply the PRN to derive coarse-grained proposals $P^\dagger = \{(s_h^\dagger, d_h^\dagger)\}_{h=1}^H$ and corresponding TRoI features $F_r$. Next, confidence scores are predicted for each proposal and regression offsets $\hat{r} = \{(\hat{s}_h, \hat{d}_h)\}_{h=1}^{H_{pos}}$ are generated to refine positive proposals, denoted as $P = \{(s_h^\dagger + \hat{s}_h, d_h^\dagger + \hat{d}_h)\}_{h=1}^{H_{pos}}$. Finally, Soft-NMS [3] is used to post-process these proposals.

## 4 Experiments

## 4.1 Experimental Setting

**Datasets.** To validate the proposed method, we conduct extensive experiments on three different datasets: LAV-DF [7], ASVS2019PS [37] (referred to as PS), and HAD [34]. Among these, the ASVS2019PS dataset is more challenging as it contains multiple and small partial forgery segments within audio.

**Compared methods.** We compare our CFPRF with existing state-of-the-art PFD methods, including PSDL [37] and IFBDN [6]. Our proposed plug-and-play PRN is integrated with these methods to realize localization results. We also adopt multi-modal TFL methods, including BA-TFD [7], BA-TFD+[5] and UMMAF [39], and adapt them to the audio TFL task by using audio-only features.

**Experiment Metrics.** To evaluate the PFD task, we report equal error rate (EER), area under the curve (AUC), false negative rate (FNR), false positive rate (FPR), and $F_1$-Score. In addition, we adopt average precision (AP) at various TIoU thresholds $\{0.5, 0.75, 0.9, 0.95\}$, average recall (AR) with different average number of proposals

**Table 1: Partial forgery detection results. Performance comparison with state-of-the-art PFD methods on evaluated datasets using different evaluation metrics.**

| Dataset | Method | EER ↓ | AUC ↑ | Pre↑ | Rec↑ | F1↑ |
|---------|--------|-------|-------|------|------|-----|
| HAD | IFBDN | 0.35 | **99.98** | 99.92 | 99.65 | 99.78 |
| | PSDL | 0.18 | 99.97 | 99.96 | 99.82 | 99.89 |
| | **CFPRF** | **0.08** | 99.96 | **99.98** | **99.92** | **99.95** |
| PS | IFBDN | 9.68 | 95.70 | 93.72 | 90.32 | 91.99 |
| | PSDL | 12.47 | 93.30 | 91.82 | 87.53 | 89.62 |
| | **CFPRF** | **7.41** | **96.97** | **95.23** | **92.59** | **93.89** |
| LAV-DF | IFBDN | 1.07 | 99.88 | 99.94 | 98.93 | 98.93 |
| | PSDL | 0.82 | **99.92** | 99.95 | 99.18 | **99.57** |
| | **CFPRF** | **0.82** | 99.89 | **99.95** | **99.18** | 99.56 |

(AN) $\{1, 2, 5, 10, 20\}$, and mean AP (mAP) under TIoU thresholds $[0.5 : 0.05 : 0.95]$ as the TFL evaluation metrics.

**Implementation Details.** We implement the proposed CFPRF using the PyTorch toolkit and conduct all experiments on the same platform with a single NVIDIA GeForce RTX 3090 GPU and the same random seed. We do not choose the experimental hyperparameters for training CFPRF detailed. The Adam optimizer is adopted to optimize the proposed CFPRF. The epoch, learning rate and weight decay are set to 30 (or 50), $10^{-7}$ (or $10^{-3}$) and $10^{-4}$ (or $10^{-3}$) respectively for training FDN (or PRN). The batch size is set to 2.

## 4.2 Comparison and Analysis

*4.2.1 Partial Forgery Detection.* We show the PFD results in Table 1. Compared to other state-of-the-art PFD methods, our CFPRF achieves the best results on all three datasets. We observe that the length and number of forgery segments within an audio can affect the detection performance of different models. For the challenging ASVS2019PS dataset, we achieve 7.41% EER (↓) and 93.89% $F_1$-Score, which show a relative improvement of 23.45% and 2.07% over the second-best method. Although our DAFL and BAFE modules are tailored for fine-grained forgery scenarios, they also provide incremental improvement for other datasets. For example, we achieve 0.08% EER on the HAD dataset and slightly outperform IFBDN (0.35% EER) and PSDL (0.18% EER). These results demonstrate the effectiveness of our proposed CFPRF for the audio PFD task.

*4.2.2 Temporal Forgery Localization.* We compare CFPRF with state-of-the-art TFL methods and PFD methods incorporated with the proposed PRN. The results are shown in Table 2, which indicates that our method outperforms the comparison methods on all three datasets in terms of mAP. As the difficulty of datasets increases, a variable decline in localization performance is observed. Specifically, our CFPRF achieves mAP scores of 99.23%, 93.01%, and 55.22% and AR@20 scores of 99.38%, 93.51%, and 66.53% on the HAD, LAV-DF, and ASVS2019PS datasets, respectively. In addition, the results achieved by TFL methods indicate the importance of feature encoders and localization headers used for the audio-only evaluation. For example, the BA-TFD and BA-TFD+ can not achieve satisfactory results due to a lack of specialized design for audio features and an unsuitable localization header BMN. While the UMMAF performs well on the datasets (HAD, LAV-DF), where audio

**Table 2: Temporal forgery localization result. Performance comparison with state-of-the-art TFL methods and PFD models integrated with PRN on evaluated datasets using different evaluation metrics, where PRN$^\dagger$ indicates coarse-grained proposals.**

| Dataset | Method | AP@0.5 | AP@0.75 | AP@0.9 | AP@0.95 | mAP | AR@1 | AR@2 | AR@5 | AR@10 | AR@20 |
|---------|--------|--------|---------|--------|---------|-----|------|------|------|-------|-------|
| HAD | BA-TFD | 79.86 | 37.98 | 5.55 | 0.57 | 40.93 | 45.12 | 47.53 | 49.99 | 52.09 | 55.15 |
| | BA-TFD+ | 88.26 | 70.69 | 37.83 | 7.39 | 64.83 | 67.49 | 68.44 | 69.06 | 69.39 | 70.15 |
| | UMMAF | **99.98** | **99.86** | 98.01 | 88.17 | 98.49 | 98.68 | 98.73 | 98.84 | 98.85 | 98.86 |
| | IFBDN+PRN$^\dagger$ | 93.85 | 91.55 | 87.75 | 79.08 | 90.40 | 96.07 | 97.39 | 97.54 | 97.54 | 97.54 |
| | IFBDN+PRN | 99.30 | 98.02 | 95.30 | 89.47 | 97.12 | 97.08 | 97.53 | 97.53 | 97.53 | 97.53 |
| | PSDL+PRN$^\dagger$ | 88.53 | 85.27 | 80.80 | 73.25 | 84.25 | 93.40 | 96.30 | 96.89 | 96.94 | 96.94 |
| | PSDL+PRN | 98.94 | 97.10 | 93.06 | 86.13 | 95.88 | 96.48 | 96.61 | 96.61 | 96.61 | 96.61 |
| | **CFPRF** | 99.77 | 99.60 | **99.21** | **96.03** | **99.23** | **99.31** | **99.38** | **99.38** | **99.38** | **99.38** |
| PS | BA-TFD | 13.65 | 4.91 | 1.06 | 0.63 | 6.15 | 8.04 | 11.03 | 15.41 | 19.14 | 23.64 |
| | BA-TFD+ | 15.72 | 6.37 | 2.05 | 1.95 | 7.69 | 7.93 | 12.62 | 18.28 | 22.17 | 26.71 |
| | UMMAF | 52.99 | 31.89 | 17.69 | 9.04 | 33.09 | 17.37 | 28.49 | 39.57 | 47.55 | 55.53 |
| | IFBDN+PRN$^\dagger$ | 43.84 | 34.79 | 27.10 | 22.53 | 34.92 | 18.72 | 33.30 | 53.87 | 60.99 | 62.22 |
| | IFBDN+PRN | 58.65 | 49.30 | 41.39 | 35.33 | 48.79 | 18.52 | 34.77 | 55.41 | 64.47 | 62.23 |
| | PSDL+PRN$^\dagger$ | 46.63 | 38.19 | 31.13 | 26.94 | 38.42 | **20.22** | 35.16 | 56.86 | 64.97 | 66.52 |
| | PSDL+PRN | 54.25 | 46.47 | 40.57 | 36.70 | 46.68 | 19.70 | **36.10** | **58.09** | 65.40 | 66.51 |
| | **CFPRF** | **66.34** | **55.47** | **48.05** | **40.96** | **55.22** | 18.48 | 35.57 | 58.06 | **65.47** | **66.53** |
| LAV-DF | BA-TFD | 53.53 | 10.98 | 0.36 | 0.02 | 20.77 | 29.56 | 32.22 | 34.73 | 38.03 | 44.66 |
| | BA-TFD+ | 83.78 | 51.99 | 6.13 | 0.46 | 49.32 | 52.78 | 54.97 | 57.21 | 58.41 | 60.04 |
| | UMMAF | **97.29** | **95.67** | 89.92 | 61.97 | 92.04 | 85.67 | 91.77 | **94.89** | **95.64** | **96.14** |
| | IFBDN+PRN$^\dagger$ | 86.83 | 84.02 | 77.85 | 70.09 | 82.55 | 86.28 | 91.78 | 92.13 | 92.13 | 92.13 |
| | IFBDN+PRN | 94.02 | 92.49 | 89.42 | 84.60 | 91.69 | 86.62 | 92.12 | 92.16 | 92.16 | 92.16 |
| | PSDL+PRN$^\dagger$ | 76.10 | 71.71 | 65.16 | 57.13 | 70.43 | 84.71 | 89.14 | 89.98 | 90.03 | 90.03 |
| | PSDL+PRN | 92.76 | 90.01 | 85.95 | 78.85 | 89.02 | 85.66 | 90.03 | 90.06 | 90.06 | 90.06 |
| | **CFPRF** | 94.52 | 93.47 | **91.65** | **88.64** | **93.01** | **87.59** | **93.49** | 93.51 | 93.51 | 93.51 |

contains a single or few forgery segments with a long span. Its performance degraded when facing small- and multi-forgery segment challenges, as evidenced by the ASVS2019PS dataset. Moreover, we observe that the localization performance improves when using features derived from PFD methods that are specifically designed for the audio modality. This further demonstrates the effectiveness of our proposed PRN, as evidenced by the 39.7%↑ (21.5%↑) relative improvement in mAP scores for the IFPRN (or PSDL) method on the ASVS2019PS dataset when incorporating PRN. In this context, we achieve relative improvements of 2.17%, 1.43%, and 13.18% respectively, on the HAD, LAV-DF and PS datasets over the second-best PFD method in terms of the mAP metrics. These results underscore the robustness of the discriminative features learned from our designed FDN and the effectiveness of PRN for two-stage localization.

## 4.3 Ablation Study

*4.3.1 Impact of FDN architecture.* We first investigate the indispensability of the various components of the proposed FDN. Specifically, we remove residual blocks (RB), dual attention layers (DAL), the DAFL module, and the BAFE module to assess their extra effectiveness for the entire network, respectively. The results are shown in Table 3, where the baseline refers to the entire FDN. We observe that DAFL has the greatest impact on performance as it enables the model to discern subtle forgery differences between real and fake frames, reducing the EER values by 0.15%, 1.82%, and 0.20% for the entire network. In addition, benefiting from the extraction of global boundary information, the BAFE module reduces the EER values by

**Table 3: Ablation study of FDN with different components.**

| Architecture | HAD | | PS | | LAV-DF | |
|-------------|--------|--------|--------|--------|--------|--------|
| | EER↓ | Δ | EER↓ | Δ | EER↓ | Δ |
| Baseline=FDN | 0.08 | 0.00 | 7.41 | 0.00 | 0.82 | 0.00 |
| w/o. RB | 0.19 | +0.11 | 9.06 | +1.65 | 0.98 | +0.16 |
| w/o. DAL | 0.09 | +0.01 | 8.09 | +0.68 | 0.86 | +0.04 |
| w/o. DAFL | 0.23 | +0.15 | 9.23 | +1.82 | 1.02 | +0.20 |
| w/o. BAFE | 0.12 | +0.04 | 8.55 | +1.14 | 0.93 | +0.11 |

**Table 4: Ablation study of PRN with different components.**

| Architecture | HAD | | PS | | LAV-DF | |
|-------------|--------|--------|--------|--------|--------|--------|
| | mAP↑ | Δ | mAP↑ | Δ | mAP↑ | Δ |
| Baseline=PRN | 99.23 | 0.00 | 55.22 | 0.00 | 93.01 | 0.00 |
| PRN → BMN | 61.92 | -37.31 | 12.35 | -42.87 | 45.61 | -47.4 |
| w/o. VegH | 98.50 | -0.73 | 38.77 | -16.45 | 86.83 | -6.18 |
| w/o. RegH | 99.11 | -0.12 | 51.76 | -3.46 | 92.90 | -0.11 |
| w/o. RB-PE | 98.39 | -0.84 | 50.96 | -4.26 | 91.52 | -1.49 |

0.04%, 1.14% and 0.11%, respectively. All in all, each component of the FDN serves a significant purpose and is thoughtfully designed.

*4.3.2 Impact of PRN architecture.* Additionally, we replace PRN with a one-stage localization header BMN and remove residual blocks of pooling encoder (RB-PE) and dual prediction headers (VegH and RegH) within the proposed PRN to show their individual effects on the localization performance. As shown in Table 4, it is

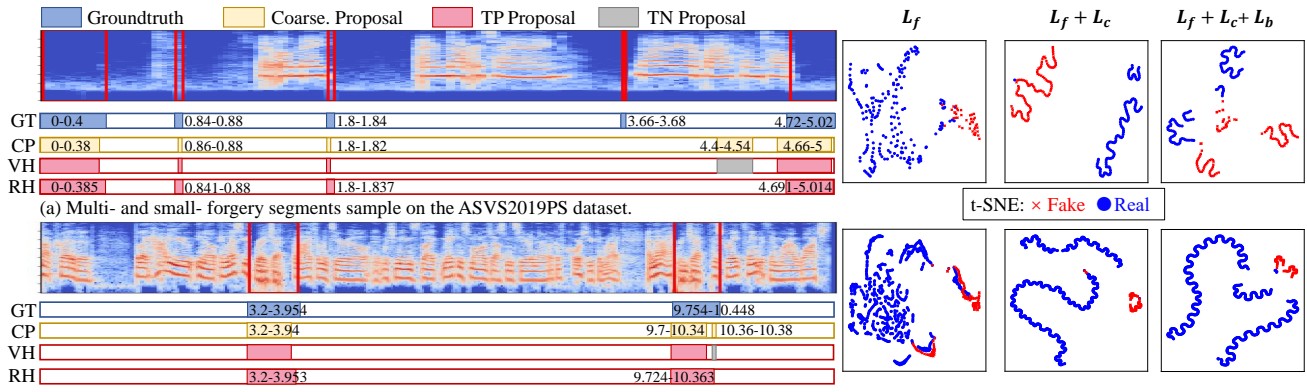

(a) Multi- and small- forgery segments sample on the ASVS2019PS dataset.

(b) Long-duration forgery audio sample on the LAV-DF dataset.

**Figure 3: Qualitative examples of our proposed CFPRF ablation experiments. From left to right: temporal forgery localization results with different PRN components and t-SNE results on the corresponding sample with different loss functions.**

straightforward to observe that replacing PRN with BMN greatly weakens the localization performance, indicating the two-stage PRN is more suitable for the audio TFL task. In addition, the TIoU features extracted from the pooling encoder that deploys RB-PE before the pooling layer can provide global context information to facilitate the identification of the verification header, thus giving better performance. Moreover, the dual prediction headers play a crucial role in the refinement of coarse-grained proposals, where VegH evaluates the completeness of these proposals, giving a 16.45% absolute improvement in the mAP score on the ASVS2019PS dataset. As shown in Figure 3, true negative proposals can be removed by the predicted confidence scores from VegH, and the RegH refines positive proposals by regression to approach groundtruths, giving a 3.46% absolute improvement in the mAP score on the ASVS2019PS dataset. In short, these components are essential to the PRN.

*4.3.3 Impact of loss functions.* To explore the contribution of each loss function in guiding frame-level feature learning, we train three models with different combinations and use t-SNE [27] to visualize the last hidden states on the different dataset samples. As shown in Figure 3, the results reveal the distinct feature distributions under different loss integrations. Specifically, the p2sgard-based MSE loss can alleviate the bias of the detection model towards classifying fake frames as real. The introduction of the CRL loss causes the feature distributions of the real and fake frames to diverge, with only a few dots intermixed at the boundary. Finally, with the inclusion of the boundary loss, we find a markedly clearer separation between the real and fake frames, as well as the fake frames within different fake regions are also clearly distinguished. This suggests that boundary information can provide global cues to enhance the detection of multiple forgery segments within an audio.

*4.3.4 Impact on model efficiency.* Finally, to compute the model efficiency, we start the experiment with 50 warm-up rounds to stabilize GPU performance, followed by 300 iterations for processing a 10-second audio. We repeat this five times and report the average results in terms of frames per second (FPS) and model parameters (#Params) in Table 5. Specifically, the FDN is supported by 320M parameters and achieves an FPS of 12270 for a 10-second audio (500

**Table 5: Complexity and inference speed of the CFPRF**

| Framework | | FPS | #Params |
|---|---|---|---|
| CFPRF | FDN | 12270 | 320M |
| | PRN | 701445 | 129K |

frames), suggesting a moderate speed and a high complexity that can potentially capture more discriminative features. In addition, the PRN utilizes 129K parameters and shows a high FPS of 701445, demonstrating its efficiency for plug-and-play localization header.

## 5 Conclusion

In this paper, we introduced a novel two-step framework called CFPRF, which incorporates an FDN and a PRN to address the emerging audio TFDL challenge. Following a two-step strategy, we first devised DAFL and BAFE modules in the FDN to mine temporal inconsistency cues caused by small and multiple forgery segments to learn robustness representations that distinguish between fake and real frames. Then, in the PRN, we derived coarse-grained proposals and region features from the FDN outputs and subsequently employed dual prediction headers to generate fine-grained proposals. Extensive experiments demonstrate that existing PFD models can achieve satisfactory localization performance with our designed PRN. Moreover, on the challenging ASVS2019PS dataset, our CFPRF could achieve a reduction of 23.45% EER for the PFD task and an improvement of 13.18% mAP for the TFL task over the second-best PFD comparative method, respectively. Finally, it is crucial to locate partial forgery segments modified by various types to provide explainability, which is the focus of our next research efforts.

## Acknowledgments

This work is supported by the National Natural Science Foundation of China (No. U2001202, No. 62072480, No. U23A20305, and No. 62172435), the Guangdong Provincial Key Laboratory of Information Security Technology (No. 2023B1212060026), and the Open Research Project of the State Key Laboratory of Media Convergence and Communication (Communication University of China) (No.SKLMCC2022KF003).

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
