# OpenReview forum: "Coarse-to-Fine Proposal Refinement Framework For Audio Temporal Forgery Detection and Localization"
_acmmm.org/ACMMM/2024/Conference — MM2024 Oral_

### Official Review · Reviewer_ygEw · 2024-05-18

**Rating:** 4
**Confidence:** 4

**Summary:**

The paper introduces a novel Coarse-to-Fine Proposal Refinement Framework (CFPRF) designed for audio temporal forgery detection and localization. It aims to improve upon existing methods by addressing the challenges of detecting subtle forgery manipulations within audio clips, particularly focusing on the accurate identification of start and end timestamps of these forgeries. The framework utilizes a Frame-level Detection Network (FDN) and a Proposal Refinement Network (PRN), incorporating techniques like difference-aware feature learning (DAFL) and boundary-aware feature enhancement (BAFE). Extensive experimental results demonstrate that the CFPRF achieves state-of-the-art performance on various datasets, including LAV-DF, ASVS2019PS, and HAD.

**Strengths:**

1. The paper introduces a novel two-stage framework for audio Temporal Forgery Localization (TFL). The two-stage model demonstrates significant performance improvements over traditional single-stage models, providing a new direction for future research in TFL tasks.
2. The FDN effectively learns robust discriminative features to mine inconsistency cues, making good use of contrastive learning techniques.
3. Extensive experiments on multiple datasets (LAV-DF, ASVS2019PS, and HAD) demonstrate the effectiveness of the proposed framework.
4. The paper is well-structured and clearly written, with each component of the proposed model thoroughly explained.
5. Figures clearly represent the algorithm's pipeline and the challenges it addresses, enhancing reader comprehension.

**Limitations:**

1. The proposed PRN network is relatively simple and lacks innovative design. In the experimental section, it may be beneficial to replace the PRN with other existing single-stage TFL or Action Localization, Sound Event Localization models. This would allow for testing the performance of the Frame-level Detection Network (FDN) and demonstrate the innovative value of the FDN.
2. The related work mentions UMMAFormer, which has already conducted experiments on audio partial forgery localization. However, the summary of this related work has some issues. Additionally, there is no performance comparison with UMMAFormer, nor have experiments been conducted on the Psynd dataset used by UMMAFormer.
3. The paper describes three types of forgery scenarios: replacement, deletion, and insertion. However, there is no evaluation or testing of the performance in these specific scenarios. For instance, it is important to investigate whether the deletion scenario might cause the FDN to fail due to the absence of significant anomalies in the feature expression, as deletion results in zero values.
4. The paper does not explain why the wav2vec2 model was chosen, nor does it provide the rationale and basis for this choice compared to other common feature extractors, including spectrogram-based or self-supervised pretrained models. It is necessary to explain the reasons and basis for choosing the wav2vec2 model and discuss its performance advantages over other common feature extractors.

**Suitability:**

2

---

### Official Review · Reviewer_PVKd · 2024-05-20

**Rating:** 5
**Confidence:** 3

**Summary:**

The paper introduces a Coarse-to-Fine Proposal Refinement Framework (CFPRF) for detecting and localizing temporal audio forgeries. It addresses the challenge of identifying subtle manipulation within long audio files, where existing methods mainly classify authenticity but fail to figure out forgery timestamps precisely. Experiments show that the proposed method achieves optimal performance in both detecting and precisely localizing audio forgeries.

**Strengths:**

1. This paper is characterized by excellent writing, clear logic, substantial content, and detailed exposition.
2. The paper proposes an effective Coarse-to-Fine Proposal Refinement Framework (CFPRF) for the challenge of identifying subtle manipulation within long audio files, in the task of detecting and localizing temporal audio forgeries.
3. Extensive experiments validate the effectiveness of the proposed CFPRF.

**Limitations:**

1. Figure 2 comprises too many elements, resulting in overlapping and obscuring, which impairs both the aesthetics and readability of the structural diagram.
2. It is better to add the up or down arrows to the all "EER" and "mAP" symbol in Table 3 and Table 4, not only when they first appear.
3. Considering the forgery detection and localization is a task needs practical application, I am curious about the model complexity and inference speed comparison of the proposed method.

**Suitability:**

2

---

### Official Review · Reviewer_teAr · 2024-05-22

**Rating:** 4
**Confidence:** 2

**Summary:**

This paper introduces a novel framework (CFPRF) to detect and localize partial forgeries in audio data. The framework addresses challenges in current countermeasures that fail to analyze the start and end timestamps of forgery segments. CFPRF includes two main components: a Frame-level Detection Network (FDN) and a Proposal Refinement Network (PRN).

**Strengths:**

• The paper effectively tackles several key challenges in audio forgery detection, such as the detection of small forgery segments, and the localization of multiple forgery segments.
 • The proposed method allows for precise localization of forgery segments. This detailed approach addresses the limitation of existing methods that mainly focus on classification without precise timestamp analysis.
 • The PRN component is designed to be plug-and-play. This flexibility enhances the framework’s applicability and ease of adoption.

**Limitations:**

• The framework requires setting certain thresholds, and the module is the combination of some previous methods.
 • Some parts and labels in the figure need to be clarified, which is hard to understand.
 • The proposed framework requires a two-step training process. This might be a constraint for rapid development and deployment, particularly in dynamic scenarios.

**Suitability:**

3

---

### Meta-Review · Area_Chair_WooM · 2024-07-02

**Recommendation:** Accept (Oral)
**Confidence:** 5

**Metareview:**

The initial ratings are 1 WA and 2 BA, and they are 3 Accept after rebuttal. The reviewers all appreciated excellent writing, clear logic, substantial content, detailed exposition, and are satisfied with the authors' responses.
The response to reviewers should be added in the final version.
I agree with the reviewers and recommend an acceptance.